# 'God protects us from death through faith and science': a qualitative study on the role of faith leaders in combating the COVID-19 pandemic and in building COVID-19 vaccine trust in Addis Ababa, Ethiopia

Kalkidan Yibeltal [1], Firehiwot Workneh [2], Hanna Melesse,[2] Habtamu Wolde,[3] Workagegnhu Tarekegn Kidane,[4] Yemane Berhane,[2] Sibylle Herzig van Wees [5]

For numbered affiliations see end of article.

**Correspondence to**
Dr Sibylle Herzig van Wees;
sibylle.hvw@ki.se

## ABSTRACT

**Objective** This study explored faith leaders' perspectives on the COVID-19 vaccine and their role in building COVID-19 vaccine trust in Addis Ababa, Ethiopia.

**Design** A qualitative study with in-depth interviews and thematic analysis was conducted.

**Participants** Twenty-one faith leaders from the seven religious groups represented in the Inter-Religious Council of Ethiopia participated in the study.

**Setting** The study was conducted in Addis Ababa, Ethiopia.

**Results** The thematic analysis revealed three themes. First, faith leaders were aware of the risks of the COVID-19 pandemic, although most ascribed a spiritual meaning to the advent of the pandemic. The pandemic seriously affected the faith communities, inflicting financial losses. Second, faith leaders were essential allies during the pandemic by effectively collaborating with government and health professionals in COVID-19 prevention activities and public health interventions using spiritual reasoning. They were actively informing the community about the importance of the COVID-19 vaccine, where many faith leaders were publicly vaccinated to build trust in the vaccine and act as role models. Third, despite this, they faced multiple questions from the congregation about the vaccine, including rumours.

**Conclusions** This research showed that faith leaders played crucial roles in encouraging vaccine use but were limited in their persuasion power because of intense rumours and misinformation. Empowering faith leaders with the latest vaccine evidence needs to be prioritised in the future.

## BACKGROUND

Equitable access to safe and effective vaccines is critical to control the spread of infectious pandemics such as COVID-19 and minimise the devastating adverse health outcomes.[1] Achieving herd immunity using effective vaccines is essential to avert pandemic-related

## STRENGTHS AND LIMITATIONS OF THIS STUDY

⇒ Prolonged engagement with data was ensured as the co-authors conducted the interviews.
⇒ Working with multiple coders aided in minimising the subjective bias of any researchers.
⇒ Interviews were conducted among faith leaders included in the structured inter-religious council; the study did not interview those outside of this structure to capture their perception and role.

morbidity and mortality.[2 3] Promoting the uptake of COVID-19 vaccines requires people's willingness to get vaccinated and ensuring proper communication of information through trusted sources to aid vaccine acceptance.[4 5]

Ethiopia started the COVID-19 vaccination campaign in March 2021 with the Astra Zeneca vaccine, manufactured by the Serum Institute of India and supplied by the COVID-19 Vaccines Global Access initiative.[6] The Ministry of Health launched a national ceremony to initiate the COVID-19 vaccine rollout in the presence of different stakeholders, including religious representatives. Initially, healthcare providers and older people with comorbidities were prioritised and given two doses of AstraZeneca vaccines.[6 7] In the later stage, the Johnson & Johnson vaccine, Sinopharm, and Pfizer vaccines were included in the campaign as they became available.[7 8]

Vaccines need to be effectively used by the target population to bring about the intended effect of controlling the pandemic. However, despite the availability of vaccines and vaccination services, vaccine hesitancy—refusal

or delay to accept a vaccine, has become a significant concern.[9 10] WHO considers vaccine hesitancy as one of the top 10 Global Health threats.[11] Concerns about vaccine safety, efficacy and side effects have driven vaccine hesitancy related to COVID-19 worldwide.[12 13] Moreover, the slow rollout and politicisation of vaccines in Africa have fuelled rumours, doubts and conspiracy theories about these vaccines.[14 15] Lack of trust in the government and the private sector, reading misinformation on social media and the history of vaccination programmes in a country have been further reasons for vaccine hesitancy among healthcare workers and the population in Africa.[13 14 16 17] Studies have shown high COVID-19 vaccine hesitancy among the Ethiopian population and some factors mentioned to explain the hesitancy were having social media as a primary information source, fear of side effects of the vaccine and thinking of it as a biological weapon and having doubts about the vaccine.[15 18–20]

Global surveys show that religious factors are the third most commonly reported reason for vaccine hesitancy globally.[21] Religious views can affect how people react to vaccinations, causing reactions like vaccine hesitancy despite medically sound and scientifically proven information/evidence.[22] In the context of the COVID-19 pandemic, concerns have been raised that religious communities may serve as the basis for misinformation and unfounded theories that undermine the use of COVID-19 vaccines.[23] Thus, the importance of the role of faith leaders in addressing vaccine hesitancy has received increasing attention in global health research.[24]

Ethiopia is a religious country where more than 97% of the population is religious, and faith leaders hold significant societal positions.[25] It has been well documented that faith leaders are embedded in society with high-influencing positions and often have considerable leverage with state and non-state actors due to the size of their constituencies.[26–29] Therefore, engagement with faith leaders is critical as they are known to be gatekeepers to local communities, with considerable influence on their communities' beliefs and behaviours. Thus, this study aimed to explore faith leaders' perspectives on the COVID-19 vaccine and the role faith leaders can play in building COVID-19 vaccine trust.

## METHODS
### Study setting
This study was conducted in Addis Ababa, the capital city of Ethiopia. Orthodox Christianity, Islam and Protestantism are the main religions in Ethiopia.[25] The Inter-Religious Council of Ethiopia (IRCE) was established in 2010 to promote inter-religious harmony in Ethiopia.[30] IRCE is represented at federal, regional and district levels. The IRCE head office is located in Addis Ababa. This council comprises seven religious groups, each with a representative in the head office.

### Study design and aim
A qualitative study design using in-depth interviews (IDI) was used to explore faith leaders' perceptions of the COVID-19 vaccine and the role faith leaders can play in building COVID-19 vaccine trust.

### Study population
The participants of this study are faith leaders selected from the seven religious groups represented in the IRCE, which comprises Ethiopian Orthodox Tewahido Church, Islam, seventh Day Adventist, Evangelical Churches Fellowship of Ethiopia, Ethiopian Kale Heywet Church, Ethiopian Evangelical Church Mekane Eyesus, and Ethiopian Catholic Church.

### Study sample and sampling procedure
Purposive sampling was used to identify appropriate study participants. The purpose of reaching out to the inter-religious council was with the notion that all the country's major religions were represented, and interviewing these participants would provide us with a comprehensive understanding of the country's context. We identified the seven religious entities constituting the IRCE through the inter-religious council. From these seven religions, individuals were selected in consultation with the head office representatives of each religion. We selected three study respondents from each religious entity: two faith leaders at the head offices and one from the religious institutions (Churches and Mosques), culminating in 21 participants. The later interviews in the study did not generate new information, so we assumed reaching data saturation. The data collection was conducted over a period of 2 months.

### Data collection procedures
IDIs were used for data collection. The interview guide was developed based on our background research. Themes included faith leaders' awareness of the COVID-19 and its risks, rumours and uncertainty around the COVID-19 vaccine, and the role of faith leaders in controlling the pandemic. The research team developed the interview guide in English and translated it into Amharic (the official language). Two research assistants with extensive experience in qualitative data collection were recruited to collect the data. The research assistants were trained for 2 days on the interview guides, the purpose of the study and research ethics. Verbal consent was obtained before the interview. All interviews were audio recorded, and additional notes were taken during the IDI. The IDIs were conducted in a private and quiet place, mainly in an area preferred by the participants, usually their offices. The interviews took 40–90 min and were conducted on dates and times convenient to the participants. COVID-19 prevention methods such as keeping physical distance, using hand sanitisers and

wearing masks were strictly implemented during the data gathering.

## Data analysis

The research assistants transcribed all interviews verbatim and then translated from Amharic to English. Transcripts and translations were checked for accuracy and consistency by two independent persons in the research group who did not conduct the interviews. Field notes taken during the IDIs were integrated into the transcriptions. KY, FW and SHvW conducted a thematic qualitative analysis following Braun and Clarke's method.[31] All data were coded double blinded. After one round of coding, the coding team met to develop a codebook and kept all audit trail records, such as data reduction notes. The codebooks included definitions and example responses. A further round of coding, using the codebook, was completed. The coding process was continued until the data were exhausted. Two further meetings were held to compare categories. Three more meetings, both in person and virtually, were made to create the themes initially by the coding team and then by the full team. Careful coding and multiple meetings ensured consistency and validity of the data analysis process. Working with multiple coders aided in minimising the subjective bias of researchers. The thematic analysis was done using Atlas.ti software V.7.5.16.

## Patient and public involvement

Patients or the public were not involved.

## Ethical considerations

Participation in this study was entirely voluntary, and informed consent was obtained from all participants. The study procedures, benefits, anticipated risks or discomforts of participating and right to withdraw were explained. Principles of anonymity and confidentiality were applied to protect the participants' identities by assigning codes to them. The audio recordings were destroyed after the completion and verification of the transcription and translations. The study was conducted following national ethical standards and the declaration of Helsinki.[32]

## RESULTS

The thematic analysis revealed three themes and 15 categories, table 1. A category was created when many codes or data extracts described that specific finding. We only use one data extract to describe the category when presenting the data in this section.

### Faith leaders were aware of the risks of COVID-19, and they and their congregations were seriously affected by the pandemic

Although most faith leaders in this study ascribed a spiritual meaning to the advent of the COVID-19 pandemic, all faith leaders interviewed were aware of the risk and

**Table 1** Themes and defined categories based on thematic analysis

| Theme | Category |
|---|---|
| Faith leaders were aware of the risks of the COVID-19 pandemic, and they and their congregations were seriously affected by the pandemic | Faith leaders were aware of the risks of the COVID-19 pandemic |
| | Solitude in difficult times |
| | Financial loss for faith institutions |
| | Vulnerable groups are left behind |
| | Spiritual crisis |
| Faith leaders are important allies during a pandemic | Faith leaders were trusted |
| | Faith leaders collaborated with government and health professionals |
| | Faith leaders engaged in public health prevention using spiritual reasoning |
| | Faith leaders trusted medical science |
| | Faith leaders were actively engaged in COVID-19 prevention methods |
| Faith leaders are faced with rumours and uncertainty around COVID-19 vaccine | Faith leaders trusted vaccines |
| | Rumours about vaccines were rampant in the community |
| | Spiritual rumours: 666 |
| | Faith leaders tried to address concerns |
| | Faith leaders were unsure how to respond to rumours |

severity of the COVID-19 pandemic. They were well informed and described in great detail the severe effects it had on their religious practice and the members of their spiritual community. In the words of a faith leader: *There is nothing it did not touch; it has affected our social life, economy, religious relations, our office, our work… We have lost so many people.* (IDI-12)

Faith leaders described the implications of solitude under challenging times, particularly during times of mourning and grief. Participants expressed profound sadness that they were unable to console individuals and support them during times of grief: *We could not even comfort the families of the deceased after the funeral. We consider spiritual support as a psychosocial treatment, but we failed to offer that too.* (IDI-4)

Moreover, faith leaders described a spiritual crisis in their institutions whereby relationships between faith leaders and those attending religious institutions had been harmed. This is mainly due to not allowing people to participate in religious services, which led to conflicts in several religious institutions. Faith leaders described a change in spiritual practice and organisation, which has remained in practice:

It [COVID-19] even created another culture; older people and people with some medical conditions became too scared to come to our congregation even after we were allowed to gather again. In our religion, Christianity, the New Testament

encourages and teaches togetherness to grow in holiness. So, it affected that core value. (IDI-6)

Faith leaders further described the financial implications for their institutions. Due to the loss of participants in religious services, income diminished substantially. Businesses surrounding religious institutions were further negatively affected. This led to significant staff cuts at religious institutions. Moreover, it led to the closure of social activities and programmes used to serve the needs of the poorest: *With the limited money we got, we were only able to pay salaries 2for Church ministers; there was nothing left to give to those who are in need.* (IDI-4)

Despite these challenges, faith leaders described innovative approaches to countering challenges. This includes the use of social media to reach the community:

We tried to solve this and reach our members through technology by calling them over the phone and using the Zoom application. Fortunately, we launched a Television programme a month or two before the pandemic called Hope Channel, and we used that to preach and praise the Lord… (IDI-14)

Active fundraising in the community, including collecting donations from individuals and businesses in the area, was further organised to make up for the financial loss:

When everyone was told to stay home, it was hard for our community because they had no money or food to eat, especially during fasting. So, we have tried our best to provide them food for the fasting period. We did that here in Addis Ababa and in some other areas, too. For that, we have collected over 30 million birr from investors and distributed it to others. (IDI-16)

### Faith leaders are important allies during a pandemic

The second theme summarises faith leaders and their institutions' work during the pandemic. To begin with, most faith leaders collaborated closely with the Ministry of Health and health professionals to inform their congregations about the disease and preventive measures. The collaboration between the government and religious institutions appears to have been effective, and faith-leaders trusted the information and directives of the Ministry of Health:

In our Church, I invited different medical doctors to convey health messages about getting vaccinated and safety measures practices recommended by the Health Ministry and the government. Based on these messages, I encourage people to practice what they have heard. (IDI-20)

In turn, faith leaders consider themselves highly trusted within their communities and congregations. They are consulted in times of hardship and on important family matters.

Religious leaders have many roles not only in controlling pandemics but also in other national affairs. This is because many people trust religious leaders; they obey them. Their word is associated with God, and what they say will get higher acceptance. Instead of a soldier speaking with a weapon, a priest holding a cross is more heard and accepted. This is a well-known fact in our country (IDI-2).

They were also consulted about questions regarding the COVID-19 pandemic, mainly rumours about the origin of the pandemic and vaccines. There are numerous examples of how faith leaders describe following directives, implementing recommended interventions (hand washing, mask use, social distancing) and actively informing the community about the importance of the COVID-19 vaccine.

Questions are forwarded to me; we often convey a message when we finish teaching in the Church. That people should follow the instructions. In general, we tell them God loves us as we obey, wash our hands, we should respect each other, and the vaccine is also important you should take it. What we do here is critical; we are spiritual doctors. (IDI-3)

Faith leaders saw themselves as key in implementing measures to address the pandemic and proactively raising awareness to help implement directives.

We intensively created awareness, stressing that the vaccine is one of the major remedies for this disease. We assigned responsible persons and Imams who can transmit these messages during the five prayer times in every Mosque. Regarding awareness creation, we successfully changed people's understanding from wrong to right direction. (IDI-16)

Faith leaders described how they used religious scriptures to promote guidelines to reduce disease transmission. This was particularly important when the community challenged them regarding these measures.

The Bible supports all the recommended preventive safety measures for COVID-19. So, if parishioners come with such concerns, I will teach them this according to the Bible and guide them in the right direction (IDI-2)

Several faith leaders, regardless of the denomination, described examples of how they used spiritual reasoning to address concerns: *…the Bible says that Luke, the Evangelist, was one of the writers of the Bible who had medical knowledge. Therefore, the doctrine of the Church does not contradict such wisdom* (IDI-11). Almost all faith leaders in this study described a strong trust in science and use religious reasoning to support this. In the words of a faith leader*: God protects us from death through faith and science* (IDI-21)

## Faith leaders are faced with rumours and uncertainty around the COVID-19 vaccine

A significant challenge described by faith leaders was the rumours about the COVID-19 vaccine they encountered. This section summarises faith leaders' different engagements with these rumours. Despite the strong commitment of faith leaders to implement preventative measures, they faced challenges in addressing rumours about vaccines. Most, but not all, faith leaders in this study trust the COVID-19 vaccine. They described their trust in science and the government and used religious reasoning to support their trust. Many faith leaders were publicly vaccinated to build trust in the vaccine: *I and other faith-leaders of different religions in Addis Ababa got vaccinated in public; we did that to be role models to our parishioners.* (IDI-13)

Participants described numerous and consistent rumours encountered in the community. The COVID-19 vaccine was considered dangerous because it was sent from abroad and was, therefore, not trusted: *Now Ethiopia is in disagreement with America as we all know, what if they send this vaccine to destroy us and attack us?"* (IDI-14). There were rumours that the COVID-19 vaccine represented 666—the mark of the beast.

There is a religious teaching about 666 (According to most manuscripts of the New Testament and in-English translations of the Bible, the number of the beast is six hundred sixty-six. (Book of Revelation 13:15–18, Wikipedia)). I guess you have heard about that in every Church. So, they say this vaccine is used to insert satanic 666 into our bodies. So, they say Satan makes the vaccine, so this is satanic teaching. They say participating in this is like partnering with Satan. (IDI-13)

Moreover, a repeated concern that faith leaders faced, and some addressed themselves, was the effect of the COVID-19 vaccine on fertility. Faith leaders described how rumours affected their own beliefs, and they described challenges in addressing these questions:

…it is rumored that getting vaccinated [against COVID-19] can cause infertility. They also say that developed countries fabricate this disease to minimize the population of highly populated countries like African countries. There were so many questions. It was tough to answer such questions. (IDI-16)

Few faith leaders actively countered the rumours, but those who did use spiritual reasoning to do so. For example:

When they say it will make you infertile, I will say, are you a prophet? Have you been given the gift of prophesying? So this shows an awareness problem due to a lack of awareness; that's how I see it. (IDI-3)

However, several faith leaders described that they got uncertain about the vaccine as a result of the rumours: *It's best to try to understand what they want to say. I don't have*

*evidence it might be what they said or it may not. So the main thing is believing.* (IDI-21)

Moreover, some described that they could not address those concerns. They were more comfortable preaching about all other public health prevention measures than the vaccine.

One of our [faith leaders'] tasks is to educate the community on preventing these outbreaks as health professionals recommended. Teaching about the disease/ pandemic is comfortable, but I cannot say anything about specific vaccines (IDI-11)

## DISCUSSION

This qualitative study explored faith leaders' perspectives towards the COVID-19 vaccine and their role in combating the pandemic and in building COVID-19 vaccine trust in Addis Ababa. The findings of our study showed that most faith leaders included in this study were aware of the risks associated with COVID-19. Due to the advent of the COVID-19 pandemic, faith leaders and their congregations have faced solitude in challenging times. Additionally, religious institutions have encountered significant financial and spiritual crises. Our study also revealed that faith leaders were actively engaged in pandemic measures and held strong beliefs in the medical sciences, including vaccines. However, faith leaders struggled to address COVID-19 vaccine rumours in their communities despite the high trust in vaccines and medical science.

This study indicated that faith leaders were well informed about the risks of COVID-19. This could be due to reasonable access to sources regarding the COVID-19 pandemic due to their networks, mass media and social media exposure. Our findings showed that due to the COVID-19 pandemic, vulnerable groups, faith leaders and their congregations have faced solitude during challenging times and significant financial and spiritual crises. This finding aligns with a qualitative study conducted in Ghana, which showed that the COVID-19 pandemic caused economic challenges to the congregants, a decline in spiritual life and a loss of fellowship and community.[33] In another qualitative study conducted on United Methodist pastors in one of the two Annual Conferences in North Carolina, Pastors reported that COVID-19 fundamentally unsettled the routine works of the ministry.[34]

This study also identified innovative approaches faith leaders use to tackle the challenges. This includes using social media to reach the community and fundraising to compensate for the financial loss. A qualitative study on Ghanaian Christian Leaders also showed that as a response to the ban, most participants in their research explicitly reported that they moved at least some activities online, which included conventional activities, such as posting prayer topics and live streaming of services for audiences who wished to view them synchronously.[35] Similarly, religious authorities in Uganda played a critical role in delivering public

health messages on COVID-19 risk communication using the web.

According to a recent systematic review that assessed religious communities' role during the COVID-19 pandemic, religion has acted as an essential platform for inter-sectoral collaboration with science and government to combat COVID-19.[36] National, regional and local faith leaders have high levels of influence and community-organising capabilities. Faith leaders are effective messengers endorsing vaccination compared with other potential messengers.[28] They can help frame approaches that make them more likely to be accepted in their communities. Governments should build trust with faith leaders and integrate them into planning, decision-making and implementation at every level of their COVID-19 response.[37] The findings in this study also showed that there was a strong collaboration between the government and religious institutions, which appeared to have been effective, and faith leaders trusted the information and directives of the Ministry of Health. This might be due to the closeness of the religious institutions to each other and the government through a well-founded and organised inter-religious council that collaborates with the government on different national agendas.[30] Besides, all faith leaders interviewed for this study consider themselves highly trusted within their communities and congregations. They were consulted about questions regarding the COVID-19 pandemic, mainly rumours about the origin of the pandemic and vaccines. Due to that, most of the study participants saw themselves as critical actors in implementing the safety measures and in proactively raising awareness to help implement directives to address the pandemic. Since faith leaders are often highly influential community leaders who are 'listened to by the community members, they often underutilise the potential for catalysing change'.[24] In line with this finding, a qualitative case study conducted in Indonesia showed that the faith leaders supported the health directives designed to reduce high transmission risk.[38] All the scholars and faith leaders stated that funerals, according to the health protocols issued by the authorities, should be conducted considering potential disease transmission. Some Islamic organisations have developed their guidelines for COVID-19 by modifying the religious values of certain institutions to prevent the spread of infectious diseases.[38]

This study revealed that regardless of the denomination, almost all faith leaders who participated in this study said they trust in science and use religious scriptures and spiritual reasoning to promote guidelines to reduce the transmission of COVID-19. A study has also shown that many faith leaders see the vaccine as a message of hope. Pope Francis also indicated everyone's moral obligation to be vaccinated.[39] Our study participants added that many faith leaders were publicly vaccinated to build trust in the vaccine. This was particularly important when the community challenged them regarding these measures. In agreement with that, findings of a qualitative study conducted in Leeds, United Kingdom, also revealed that faith leaders could play a vital role in the health behaviour of their congregants.[27] Faith leaders can influence health behaviour not only on the individual but also on a sociocultural and environmental level. They exert such influence through several mediators, including as scriptural power, social influencer and serving as role models.[27] Our results showed most of the faith leaders faced numerous and consistent rumours in the community. They were affected by the rumours they heard about the COVID-19 vaccine and faced challenges in addressing such questions.

The African Center for Strategic Studies reported that the preponderance of COVID-19 vaccine myths is causing many Africans to forego vaccinations when new, more transmissible coronavirus variants spread across the continent.[40] The report indicated several myths, including live viruses are injected to cause death after vaccination; the vaccine causes infertility by altering the DNA to reduce Africa's population, and other serious side effects that are worse than contracting the virus, and some claim that vaccines are a cover to implant traceable microchips.[40] Such myths and similar rumours may lead to COVID-19 vaccine hesitancy in the general community. In line with these findings, a population-based online survey conducted in Ethiopia showed that only 31.4% of participants were willing to receive the COVID-19 vaccine.[19] One of the strategies to counter vaccine hesitancy is to follow a multisectoral approach that involves the collaboration between various stakeholders, such as government, private companies, religious groups and other agencies, to leverage the knowledge, expertise and resources, thereby enabling the creation of longstanding public trust of vaccines.[27 39] Given the importance of faith leaders and their trust in science, equipping faith leaders with the knowledge and skills to address rumours appears indispensable.

Limitations of the study can lay in the fact that we only interviewed faith leaders included in the structured inter-religious council and did not directly interview those outside of this structure to capture their perceptions and roles. Although the findings cannot be generalised, inferential generalisations can be drawn, and results may be relevant for other similar settings.

## CONCLUSION

Faith leaders have played an essential role in mitigating the COVID-19 pandemic in Ethiopia. Faith leaders have followed and implemented government directives and promoted the vaccine, further amplifying their important alliance in public health. However, they have been challenged with different rumours about the vaccine, which they struggled to address as they were not equipped to do

so. Thus, empowering faith leaders to address rumours about new vaccines could have significant public health implications for ongoing and future pandemics. Future studies need to consider involving faith leaders at all levels and to utilise quantitative methods. In addition, intervention studies addressing issues of hesitancy are necessary.

**Author affiliations**

[1]Department of Reproductive Health and Population, Addis Continental Institute of Public Health, Addis Ababa, Ethiopia

[2]Department of Epidemiology and Biostatistics, Addis Continental Institute of Public Health, Addis Ababa, Ethiopia

[3]Independent Consultant, Addis Ababa, Ethiopia

[4]Department of Nutrition and Behavioral Sciences, Addis Continental Institute of Public Health, Addis Ababa, Ethiopia

[5]Department of Global Public Health, Karolinska Institutet, Stockholm, Sweden

**Acknowledgements** We would like to acknowledge the Inter-Religious Council of Ethiopia for their collaboration for the study.

**Contributors** YB, SHvW, FW and KY designed the study. HM and HW conducted the interviews, transcriptions and translations. KY, FW, HM, WT and SHvW were involved in the coding and conducted the analysis. All authors collaborated on the writing of the manuscript, and the final draft was written by KY and FW. YB is the guarantor of this study. All authors have read and approved the final manuscript.

**Funding** The Addis Continental Institute of Public Health, Ethiopia, funded the study. SHvW is the recipient of a Postdoc grant awardee by FORTE (2021-01299) in Sweden.

**Competing interests** None declared.

**Patient and public involvement** Patients and/or the public were not involved in the design, or conduct, or reporting, or dissemination plans of this research.

**Patient consent for publication** Consent obtained directly from patient(s).

**Ethics approval** This study involves human participants and was approved by Institutional Review Board (IRB) of Addis Continental Institute of Public Health (ACIPH/IRB/010/2021). Participants gave informed consent to participate in the study before taking part.

**Provenance and peer review** Not commissioned; externally peer-reviewed.

**Data availability statement** Data are available upon reasonable request.

**ORCID iDs**

Kalkidan Yibeltal http://orcid.org/0000-0001-5882-5767

Firehiwot Workneh http://orcid.org/0000-0003-1529-987X

Sibylle Herzig van Wees http://orcid.org/0000-0002-5270-1170

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
