## [Reviewer comments · BMJ Open]

ARTICLE DETAILS

TITLE (PROVISIONAL)	“God protects us from death through faith and science”: A qualitative study on the role of the faith-leaders in combating the COVID-19 pandemic and in building COVID-19 vaccine trust in Addis Ababa, Ethiopia.
AUTHORS	Yibeltal, Kalkidan; Workneh, Firehiwot; Melesse, Hanna; Wolde, Habtamu; Kidane, Workagegnehu Tarekegn; Berhane, Yemane; Herzig van Wees, Sibylle

VERSION 1 – REVIEW

REVIEWER	Powe, Nicolette Youngstown State University, Graduate studies Health and Rehabilitation Sciences
REVIEW RETURNED	08-Jul-2023

GENERAL COMMENTS	Summary: This study aimed to explore faith leaders' perspectives on the COVID-19 vaccine and their role in building COVID-19 vaccine trust in Addis Ababa, Ethiopia. A qualitative study design using in-depth interviews was used to explore faith-leaders' perceptions of the COVID-19 vaccine and the role faith-leaders can play in building COVID-19 vaccine trust. The thematic analysis revealed three themes and 15 categories. Theme 1) Faith-leaders have a clear awareness of COVID-19, and they and their congregations have been seriously affected by the pandemic. Theme 2) Faith leaders are important allies during a pandemic. Theme 3) Faith-leaders intrigued by rumors and uncertainty around COVID-19 vaccine. Faith-leaders were effective messengers endorsing vaccination from being accepted in their communities. By building trust with faith-based leaders, governments could integrate their planning, decision making, and implementation at every level of their COVID-19 response. The findings in this study supports the utilization well-founded and organized inter-religious council that collaborates with the government on different national agendas. Faith-leaders serve as key actors in implementing the safety measures and in proactively raising awareness to help implement directives to address the pandemic. Strengths: The study clearly outlines the role that religious communities may play in combatting misinformation. This qualitative study that used in-depth interviews explored faith-leaders' perceptions of the COVID-19 vaccine and the role faith-leaders can play in building COVID-19 vaccine trust. The study findings from thematic analysis revealed three themes and 15 categories. Overall, the study supports existing literature of effective faith-based collaboration with
---

	government. Weakness:  1) Abstract Page 3 Line 19: How do you evaluate “understand”? Consider replacing “understand” and another word choice. 2) Background Page 5 Line 16: How do you evaluate “understood”? Consider replacing “understood” and another word choice. 3) Methods Page 5 Line 28-35: The breakdown provided of the main religions in Ethiopia is background content or results? It is not clear if the statistics provided was to be demographic results description or a description of participants? 4) Results Page 8 Line 46 & Page 11 Line 3: How do you evaluate “understanding”? Consider replacing “understanding” and another word choice. 5) Results Page 8 Line 52: How do you evaluate “understand”? Consider replacing “understand” and another word choice. 6) Discussion Page 13 Line 9: How do you evaluate “understand”? Consider replacing “understand” and another word choice. 7) Consider how to shorten the results and discussion section to focus on recommendations and implication of future studies.
--	--

REVIEWER	DiGregorio, Bernard D. West Virginia University, Sociology and Anthropology
REVIEW RETURNED	04-Aug-2023

GENERAL COMMENTS	In this paper, the authors used in-depth interviews to explore how faith leaders whose religious traditions were represented in the Inter-Religious Council of Ethiopia (IRCE) perceived the COVID-19 pandemic, as well as why they make valuable allies in responding to the virus, and some of the negative beliefs and arguments that they have received from their congregants. The authors themselves note that due to the non-random nature of the sampling, the findings can not be generalized to Ethiopian religious leaders as a whole. However, they are also correct in that despite this limitation, the research conducted provides us with a better look at the pandemic experience for Ethiopian religious leaders, which can in turn provide the foundation for future research. I am also glad to see research on the pandemic which focuses on a non-US/European population, further allowing us to see responses to the pandemic from a different cultural context. Some of the rumors and myths mentioned about the COVID-19 vaccine matched up with what we’ve seen in the US and Europe, but others were uniquely African. With that being said, I note two major issues (though one is a quick fix) and five minor issues that I would like to see addressed by the authors. Major issues:  1) The first major issue that I have is that what I’ve read here leaves me wanting more information. It all looks interesting, but I feel that in some cases, more information is needed, while in others, it would make for a stronger paper.  a. First and foremost, while the authors provide some quotes to support their themes, I find myself wanting a table, some numbers to look at. Are the quotes representative of the group, or are they the exception? Especially when the authors make claims such as most religious leaders ascribing a spiritual meaning to COVID-19 or most trusting the vaccine, I need more information. At the very least, how many of the respondents are “most” in each of these cases? Were there any patterns to those who didn’t treat the pandemic as
--

spiritual, or who didn't trust the vaccine? Specific religious traditions or level of authority across traditions? To simplify, I'd love a table that tells me how many religious leaders said something for each theme/category, perhaps with an indicator of which religious traditions are represented, and at what levels. The work is qualitative but can still benefit from some numbers.

b. The authors note that one of the category areas is religious leaders being trusted, but I don't recall seeing any quotes to that effect. While the table noted above would help to provide more support for this, I would ideally like to see at least one quote supporting this argument.

c. Along the same lines, one of the categories is collaboration between the religious leaders and the government, but the quote supporting this focuses on the religious leaders and people obeying the government. I would call this deference, rather than collaboration. Are there any quotes supporting active collaboration (such as meeting with government officials to get specific information to convey to their congregants, or inviting officials to talk, things of that nature)?

d. Finally, the authors note that some religious leaders didn't trust the virus, and that some also didn't counter rumors about the vaccine. While not necessary, I would love to see a bit more information here, about why they don't trust them, or don't counter rumors. The authors focus mostly on the positives of religious leaders dealing with these issues, but it wouldn't hurt to also note what issues are arising, especially if the conclusion is one which argues for helping to support religious leaders in countering these rumors and problems.

2) The second major issue that I have appears to be an issue of plagiarism. On page 4, under the study setting, I wanted to get a better idea of what was represented under traditional faiths, so I loaded up the reference noted for that sentence. In doing so, I found that the first three lines of that section were pulled almost verbatim from the source listed, with the only change being to substitute "Islam" for "Muslim" in the second line. The authors reference the source, but there are no markings indicating that this is a direct quote. Either the wording needs to be changed into the author's own words, or the proper quotation marks need to be applied. While a major issue, this is one that is much more easily addressed.

In addition to these two major issues, I also have five minor issues that I feel need to be addressed.

1) On p. 3, the authors suggest that one reason for vaccine hesitancy in Ethiopia is due to the history of vaccination there. I'm not familiar with the context, and I suspect that many readers would also be less familiar with it, so I would love to see this discussed in a bit more detail, as it is important to note.

2) On p. 4, the authors claim that it is well documented that religious leaders are embedded in society with high influencing positions. With a claim like that, I would like to see multiple citations to back up that claim. I'm not disputing it, just asking for evidence in support of it, for other readers who may be more contentious.

3) There are numerous grammatical errors and areas where rewording would be beneficial. While not a complete list, this should provide some areas to focus on:

	a. On p. 3, the sentence explaining what vaccine hesitancy is comes across as unwieldy and could be rewritten for clarity. At the end of the same paragraph, where the authors list reasons for vaccine hesitancy, mistrust is mentioned in two different arguments. I would love to see this area reworded to present each reason a single time, in order to be clearer. b. On p. 5, the study sample and sampling procedure could also be reworded, in order to be clearer about the sample makeup. c. Also on p. 5, the authors mention that two independent persons checked the translations. Independent from the interviews, or from the larger research project? d. On p. 6, at the top of the page, there is the sentence, “Working with multiple coders aid in...”. You want to keep the tense the same throughout this section, a past tense going over what work was undertaken, so you’ll want to switch “aids” to “aided”. e. Some of the grammatical errors may be found on p. 4, Inter-Religious Council of Ethiopian, rather than Ethiopia, and on p. 13, Pope Frances, rather than Francis. 4) Many of the references need to be redone, to include authors where possible. a. One example of this is reference 12, citing from BMJ Open. It does not list the authors, though following the link I was able to determine the publication information and authors. For some of the other sources, the items being referenced have suggestions on how to cite them, which could also help. 5) Finally, I propose making a couple of small changes to the title and theme areas presented in this draft. a. I would expand on the title, as the first thematic area, on faith leaders’ understanding of COVID-19, is more encompassing than just views on vaccines and vaccine trust, and the second thematic area seems to focus on faith leaders as allies in responding to pandemics in general. b. I would also shift the category of faith leaders trusting vaccines from the section on rumors and uncertainty surrounding the vaccine, to the section focusing on why faith leaders make good allies. That area seems less to do with the issues that faith leaders deal with, and instead provides further support for what would make them good allies in responding to a pandemic. While I realize that I am asking for a goodly amount of work to be done to this manuscript, I believe that it will be the stronger for it, and will make a greater contribution to our understanding of the role and relationships of faith leaders in response to a pandemic.
--	---

VERSION 1 – AUTHOR RESPONSE

Reviewer: 1

Dr. Nicolette Powe, Youngstown State University

Thank you for the constructive reviews, which we believe have improved this paper.

1) Abstract Page 3 Line 19: How do you evaluate “understand”? Consider replacing “understand” and another word choice.

Response: Thank you for indicating an important issue. We have replaced it with an appropriate word depending on the statement.

2) Background Page 5 Line 16: How do you evaluate “understood”? Consider replacing “understood” and another word choice.

Response: We have replaced it with an appropriate word for the statement.

3) Methods Page 5 Line 28-35: The breakdown provided of the main religions in Ethiopia is background content or results? It is not clear if the statistics provided was to be demographic results description or a description of participants?

Response: The information provided under the study setting is a background description of the country and the major religions in Ethiopia. We have removed the statistics to avoid confusion.

4) Results Page 8 Line 46 & Page 11 Line 3: How do you evaluate “understanding”? Consider replacing “understanding” and another word choice.

Response: We have replaced “understanding” with a more suitable term.

5) Results Page 8 Line 52: How do you evaluate “understand”? Consider replacing “understand” and another word choice.

Response: We have replaced “understand” with a fitting word.

6) Discussion Page 13 Line 9: How do you evaluate “understand”? Consider replacing “understand” and another word choice.

Response: We have replaced “understand” with a better term.

7) Consider how to shorten the results and discussion section to focus on recommendations and implication of future studies.

Response: We have tried to balance the length of the results and discussion, considering the other reviewer's request for more.

Reviewer: 2

Thank you for the constructive reviews, which we believe have improved this paper.

Major issues:

1) The first major issue that I have is that what I've read here leaves me wanting more information. It

all looks interesting, but I feel that in some cases, more information is needed, while in others, it would make for a stronger paper.

a. First and foremost, while the authors provide some quotes to support their themes, I find myself wanting a table, some numbers to look at. Are the quotes representative of the group, or are they the exception? Especially when the authors make claims such as most religious leaders ascribing a spiritual meaning to COVID-19 or most trusting the vaccine, I need more information. At the very least, how many of the respondents are “most” in each of these cases? Were there any patterns to those who didn’t treat the pandemic as spiritual, or who didn’t trust the vaccine? Specific religious traditions or level of authority across traditions? To simplify, I’d love a table that tells me how many religious leaders said something for each theme/category, perhaps with an indicator of which religious traditions are represented, and at what levels. The work is qualitative but can still benefit from some numbers.

Response: The study was not designed to produce numbers, so we applied thematic analysis as in many previously published articles on the subject matter. We refer to “most” or “many” if no differing views were expressed.

b. The authors note that one of the category areas is religious leaders being trusted, but I don’t recall seeing any quotes to that effect. While the table noted above would help to provide more support for this, I would ideally like to see at least one quote supporting this argument.

Response: The following quotes support our arguments.

“Religious leaders have many roles not only in controlling pandemics but also in other national affairs. This is because many people trust religious leaders; they obey them. Their word is associated with God, and what they say will get higher acceptance. Instead of a soldier or a policeman speaking with a weapon, a small priest holding a cross is more heard and accepted. This is a well-known fact in our country.” (ID1-2)

“Because they have God-given obligation. They sat there and the people also trusted them. They, in turn, are compelled by their religion to lead the people. This is their responsibility.” ID1-16

c. Along the same lines, one of the categories is collaboration between the religious leaders and the government, but the quote supporting this focuses on the religious leaders and people obeying the government. I would call this deference, rather than collaboration. Are there any quotes supporting active collaboration (such as meeting with government officials to get specific information to convey to their congregants or inviting officials to talk, about things of that nature)?

Response: We understand this is a bit tricky. However, according to the qualitative study coding practices, the coding team agreed on using the term collaboration; this includes respect for policies, meeting attendance, and inviting officials. The following quotes support collaboration:

“In our church, I invited different medical doctors to convey health messages about getting vaccinated and safety measure practices recommended by the health ministry and the government. Based on these messages, I encourage people to practice what they have heard.”
IDI-20

“Representatives of different religions gathered and made a decision that based on the directives given by the government, every religious entity are allowed to design their operational guideline for the solution in a manner that strengthens government effort.” IDI-20.

“Around the beginning, the government has talked with us. The government officials told us that, as the pandemic was a sudden disaster and there were no adequate ventilators and the available hospitals were also too narrow, our government is not able to save the people like other countries. So we accepted the government's offer to help in any way we could.” IDI-18

d. Finally, the authors note that some religious leaders didn't trust the virus and that some also didn't counter rumors about the vaccine. While not necessary, I would love to see a bit more information here about why they don't trust them, or don't counter rumors. The authors focus mostly on the positives of religious leaders dealing with these issues. Still, it wouldn't hurt also to note what issues are arising, especially if the conclusion is one which argues for helping to support religious leaders in countering these rumors and problems.

Response: We did not have enough data to explore this in greater detail because only a few religious leaders mentioned this.

2) The second major issue that I have appears to be an issue of plagiarism. On page 4, under the study setting, I wanted to get a better idea of what was represented under traditional faiths, so I loaded up the reference noted for that sentence. In doing so, I found that the first three lines of that section were pulled almost verbatim from the source listed, with the only change being to substitute “Islam” for “Muslim” in the second line. The authors reference the source, but there are no markings indicating that this is a direct quote. Either the wording needs to be changed into the author's own words, or the proper quotation marks need to be applied. While a major issue, this is one that is much more easily addressed.

Response: Thank you for picking up this important matter. It has been paraphrased now.

In addition to these two major issues, I also have five minor issues that I feel need to be addressed.

1) On p. 3, the authors suggest that one reason for vaccine hesitancy in Ethiopia is due to the history of vaccination there. I'm not familiar with the context, and I suspect that many readers would also be less familiar with it, so I would love to see this discussed in a bit more detail, as it is important to note.

Response: Sorry for the lack of clarity. In that paragraph, we mentioned the factors related to vaccine hesitancy in Africa. We have tried to clarify and added a statement about the Ethiopian situation.

2) On p. 4, the authors claim that it is well documented that religious leaders are embedded in society with high influencing positions. With a claim like that, I would like to see multiple citations to back up that claim. I'm not disputing it, just asking for evidence in support of it, for other readers who may be more contentious.

Response: Good point; we have added references to back the statement.

3) There are numerous grammatical errors and areas where rewording would be beneficial. While not a complete list, this should provide some areas to focus on:

a. On p. 3, the sentence explaining what vaccine hesitancy is comes across as unwieldy and could be rewritten for clarity. At the end of the same paragraph, where the authors list reasons for vaccine hesitancy, mistrust is mentioned in two different arguments. I would love to see this area reworded to present each reason a single time, in order to be clearer.

Response: We have reworded the paragraph to make it more clear.

b. On p. 5, the study sample and sampling procedure could also be reworded, in order to be clearer about the sample makeup.

Response: We have reworded the sampling procedure to improve readability.

c. Also on p. 5, the authors mention that two independent persons checked the translations. Independent from the interviews, or from the larger research project?

Response: Though part of the research team, the independent persons didn't conduct the interviews. We have now clarified it in the manuscript.

d. On p. 6, at the top of the page, there is the sentence, "Working with multiple coders aid in...". You want to keep the tense the same throughout this section, a past tense going over what work was undertaken, so you'll want to switch "aids" to "aided."

Response: This has been corrected now.

e. Some of the grammatical errors may be found on p. 4, Inter-Religious Council of Ethiopian, rather than Ethiopia, and on p. 13, Pope Frances, rather than Francis.

Response: This is corrected now.

4) Many of the references need to be redone, to include authors where possible.
a. One example of this is reference 12, citing from BMJ Open. It does not list the authors, though following the link I was able to determine the publication information and authors. For some of the other sources, the items being referenced have suggestions on how to cite them, which could also help.

Response: Thank you for picking this up. All references have been redone.

5) Finally, I propose making a couple of small changes to the title and theme areas presented in this draft.

a. I would expand on the title, as the first thematic area, on faith leaders' understanding of COVID-19, is more encompassing than just views on vaccines and vaccine trust, and the second thematic area seems to focus on faith leaders as allies in responding to pandemics in general.

Response: We have modified the title to: “God protects us from death through faith and science”: A qualitative study on the role of the faith leaders in combating the COVID-19 pandemic and in building vaccine trust in Addis Ababa, Ethiopia

b. I would also shift the category of faith leaders trusting vaccines from the section on rumors and uncertainty surrounding the vaccine to the section focusing on why faith leaders make good allies. That area seems less to do with the issues that faith leaders deal with, and instead provides further support for what would make them good allies in responding to a pandemic.

Response: Thank you for the suggestion; after careful consideration, the research team reached a consensus to keep the categories as in the initial draft.

While I realize that I am asking for a goodly amount of work to be done to this manuscript, I believe that it will be the stronger for it, and will make a greater contribution to our understanding of the role and relationships of faith leaders in response to a pandemic.

VERSION 2 – REVIEW

REVIEWER	Powe, Nicolette Youngstown State University, Graduate studies Health and Rehabilitation Sciences
REVIEW RETURNED	26-Jan-2024

GENERAL COMMENTS	Method: Page 6 Line 120; Page 6 Line 129: How do you evaluate “understand”? Consider replacing “understand” and another word choice. Methods: Page 6 Line 125: “Interview conducted later in the study” – How long did it take to conduct 21 participants to reach saturation? Methods: Page 7 Line 153: Define “several team meetings”. How many meetings did it take to develop and refine the themes? Explain what was done during the team meetings to identify themes. Results Page 9 Line 221: Are you able to provide an example of active fundraising? Results Page 10 Line 229: Define “most faith leaders”. Are you able to provide a percentage of faith leaders collaborated with Ministry of Health? Results Page 10 Line 237: Are you able to provide a percentage of faith leaders who considered themselves highly trusted? Results Page 11 Line 271: Are you able to provide a percentage of faith-leaders who described a strong trust?
--

	Results Page 11 Line 276: Are you able to provide a percentage of faith-leaders who was the rumors? Results Page 11 Line 281: Are you able to provide a percentage of faith-leaders who were publicly vaccinated? Results Page 12 Line 294-295: Are you able to described rumors show up in social media? Discussion Page 14 Line 317: How do you evaluate “understand”? Consider replacing “understand” and another word choice. Discussion Page 15 Line 344, Line 346, Line 354, Line 363, Line 366: Add space between word and “(“ or delete space before period. Discussion Page 16 Line 375, Line 378, Line 388, Line 393: Add space between word and “(“ or delete space before period. Discussion Page 16 Line 389-393: Provide clear recommendations for future studies Conclusion Page 16 Line 403-406: Provide clear recommendations for future studies
--	---

REVIEWER	DiGregorio, Bernard D. West Virginia University, Sociology and Anthropology
REVIEW RETURNED	21-Dec-2023

GENERAL COMMENTS	The authors have addressed all of the points that I raised, as well of those of the other reviewer, making changes where applicable and providing valid reasoning for those which were not changed, resulting in a stronger paper overall.
--

VERSION 2 – AUTHOR RESPONSE

Reviewer: 1

Dr. Nicolette Powe, Youngstown State University

Comments to the Author:

1. Method: Page 6 Line 120; Page 6 Line 129: How do you evaluate “understand”? Consider replacing “understand” and another word choice.

Response: Thank you, we have now replaced understand with a better term.

2. Methods: Page 6 Line 125: “Interview conducted later in the study” – How long did it take to conduct 21 participants to reach saturation?

Response: It took us two months to reach saturation (line 127).

3. Methods: Page 7 Line 153: Define “several team meetings”. How many meetings did it take to develop and refine the themes? Explain what was done during the team meetings to identify themes.

Response: We had at least five scheduled meetings to discuss the coding and the analysis process. This is explained under the data analysis section now.

4. Results Page 9 Line 221: Are you able to provide an example of active fundraising?

Response: Fundraising was primarily from members of the community and local businesses. This is indicated in the narrative (lines 223-224).

5. Results Page 10 Line 229: Define “most faith leaders”. Are you able to provide a percentage of faith leaders collaborated with Ministry of Health?

Response:As this is a qualitative study, it was not designed to provide numbers and percentages, so we applied thematic analysis as in many previously published studies on the subject matter. We refer to “most” or “many” if no differing views were expressed.

6. Results Page 10 Line 237: Are you able to provide a percentage of faith leaders who considered themselves highly trusted?

Response: See response above

7. Results Page 11 Line 271: Are you able to provide a percentage of faith-leaders who described a strong trust?

Response:See response above

8. Results Page 11 Line 276: Are you able to provide a percentage of faith-leaders who was the rumors?

Response:See response above

9. Results Page 11 Line 281: Are you able to provide a percentage of faith-leaders who were publicly vaccinated?

Response:See response above

10. Results Page 12 Line 294-295: Are you able to describe rumors show up in social media?

Response: Yes, based on responses which are presented in lines 299-302. However, we did not attempt to collect data directly from social media, as this was not the scope of this study.

11. Discussion Page 14 Line 317: How do you evaluate “understand”? Consider replacing “understand” and another word choice.

Response:Thank you, we have replaced “understand” with a fitting term.

12. Discussion Page 15 Line 344, Line 346, Line 354, Line 363, Line 366: Add space between word and “(“or delete space before period.

Response: Thank you for picking these up, all are addressed now.

13. Discussion Page 16 Line 375, Line 378, Line 388, Line 393: Add space between word and “(“or delete space before period.

Response: Thank you for picking these up, all are addressed now.

14. Discussion Page 16 Line 389-393: Provide clear recommendations for future studies

Response: We have added a recommendation for future study under the conclusion section (lines 409-410).

15. Conclusion Page 16 Line 403-406: Provide clear recommendations for future studies

Response: We have added a recommendation for future study under the conclusion section (lines 408-409).

Reviewer: 2

Dr. Bernard D. DiGregorio, West Virginia University

Comments to the Author:

The authors have addressed all of the points that I raised, as well as those of the other reviewer, making changes where applicable and providing valid reasoning for those which were not changed, resulting in a stronger paper overall.

VERSION 3 – REVIEW

REVIEWER	DiGregorio, Bernard D. West Virginia University, Sociology and Anthropology
REVIEW RETURNED	11-Mar-2024
GENERAL COMMENTS	I agree with the suggestions of the other reviewer, and believe that the revisions that the authors engaged in have strengthened the paper, particularly the change in word choice from "understand" to "awareness", which reflects knowledge without assuming what level of analysis and weighing of all factors is at play. As I have previously mentioned, I appreciate the research of the authors as providing us with an exploration of the role of religion and religious leaders in navigating the COVID-19 pandemic, outside of a US or European context.